# Exploring women's experience of healthcare use during pregnancy and childbirth to understand factors contributing to perinatal deaths in Pakistan: A qualitative study

**Jamil Ahmed**[1,2]*, **Ashraful Alam**[2], **Saadat Khokhar**[3], **Sadiq Khowaja**[4], **Ramesh Kumar**[5], **Camille Raynes Greenow**[2]

1 Department of Family and Community Medicine, College of Medicine and Medical Sciences, Arabian Gulf University, Manama, Bahrain, 2 The University of Sydney, Sydney School of Public Health, Camperdown, New South Wales, Australia, 3 JHPIEGO Pakistan-an affiliate of Johns Hopkins University, Lahore, Pakistan, 4 District Health Office Tando Muhammad Khan, Sindh, Pakistan, 5 Health Services Academy, Islamabad, Pakistan

* jahm3675@sydney.edu.au

**Data Availability Statement:** Due to participant confidentiality concerns, the qualitative data cannot

## Abstract

Understanding key healthcare system challenges experienced by women during pregnancy and birth is crucial to scale up available interventions and reduce perinatal mortality. A community perspective about preferences and experience of care during this period can be used to improve community-based programs to reduce perinatal mortality. Using a qualitative exploratory approach, we examined women's experience of perinatal loss, aiming to understand the main factors, as perceived and experienced by women, leading to perinatal loss. Qualitative in-depth Interviews were conducted with 25 mothers with a recent perinatal loss, three family members, six healthcare officials, and two focus group discussions with 17 lady health workers. Data were analysed using inductive and deductive coding, by thematic analysis. Our findings revealed three distinct but interrelated themes, which include: 1) poor access to care during pregnancy and birth, 2) unavailability of appropriate healthcare services, and 3) poor quality of care during pregnancy and birth. Women frequently delayed seeking formal care around birth because of delays by themselves, their family members, or the local traditional birth attendants who frequently induced births at women's homes without recognising the dangers to the mothers or their babies. Preference for private care was common, however they often could not bear the cost of care when they needed caesarean section or in-patient care for their sick newborns because these services were absent in public health facilities of the district. Referral to the regional tertiary care hospital was often not officially arranged leading to risky births in small and crowded private clinics. Women's views about negative staff attitudes and the lack of attention given to them in public health facilities highlighted a lack of quality and respectful antenatal care. Improvement in women's access to essential care during pregnancy and around birth, availability of emergency obstetric and newborn care, improving the quality of maternal and newborn care in both public and private health facilities at the district level might reduce perinatal mortality in Pakistan.

be made publicly available. The data can be requested from the corresponding author or from the institution that provided the authors with ethical approval: Adnan Khan, Assistant Registrar, Health Services Academy, Islamabad Pakistan, adnan. khan810@gmail.com.

**Funding:** This work was supported by Postgraduate Research Support Scheme and Research Student Grant Scheme of the University of Sydney, Australia. CRG was funded by an NHMRC CDF #1086062, and a Robinson Fellowship, University of Sydney.

**Competing interests:** The authors have declared that no competing interests exist.

## Introduction

The burden of perinatal mortality comprising of stillbirths and early newborn deaths continues to be a challenge in low-income countries particularly in South Asia and Sub-Saharan Africa [1]. Of the approximately 3.2 million global stillbirths, more than 98% occur in low and middle-income countries like Pakistan [2]. Pakistan has the third highest stillbirth rate in the world, at 47 per 1000 births, resulting in ~242, 600 fetal deaths at 28 weeks of gestation in 2015. With ~244, 700 neonatal deaths within first week after birth in 2015, Pakistan ranks second in the list of countries with the highest neonatal mortality rate [3]. To put this in perspective, most high-income countries have an average stillbirth rate of 3·5 per 1000 total births.[4].

Perinatal mortality is a highly sensitive indicator of the quality and accessibility of care received during pregnancy and at the time of birth [5]. This suggests that some perinatal mortality is preventable through interventions [6]. Countries like Bangladesh, Cambodia, and Rwanda outperformed other high burden countries, and achieved an annual reduction of more than 3.5% in stillbirths between 2000 and 2015, because they were able to improve women's access to care during pregnancy and birth [3]. In countries which lag behind, bottlenecks in intervention implementation continue to be a challenge to further reductions in perinatal mortality. For instance a bottleneck analysis for newborn survival identified wide gaps in almost all aspects of the implementation of intervention packages aimed at in-patient care of the small and sick newborns [7]. In rural populations in Pakistan, despite improved access to primary healthcare services, the quality of care still lags behind best practice [8]. Although the main risk factors [9] and interventions to prevent perinatal mortality have been identified [10], there still exist gaps in our understanding of how these interventions are integrated within community-based maternal and child healthcare services in countries with a high burden of perinatal mortality.

Understanding community experience of care around birth in Pakistan may provide evidence to reorient maternal and child healthcare policies and service delivery. In our previous study in which we analysed population-level data, we found that women with a high perinatal loss in Pakistan were also less empowered to seek care for themselves, their sick children, or use methods for birth spacing [11]. In this present study, we explored women's experience of using healthcare during pregnancy and birth to understand factors related to healthcare access and utilisation which may have contributed to their perinatal losses.

## Methodology

We used exploratory qualitative research methodology to meet the aims of this study. In-depth interviews (IDIs) and focus group discussion (FGDs) were the main methods used for data collection.

### Setting

The study was conducted in Tando Muhammad Khan District in Sindh Province, Pakistan during June and August 2018. The district was selected because it is one of the typical rural districts in the province of Sindh. The region is a predominately rural with a population of ~677,228. There are 11 mother and child health centres functioning within either basic health units, rural health centres or the district hospital. These are all public health facilities and are referred to as that throughout the paper. The district has two major private medical centres, one of which is a teaching hospital and affiliated with a private medical college. The district also has approximately 20 smaller private maternity care providers. There are 83 community midwives in the rural areas who have established stations in their villages with

initial support by the provincial Maternal and Child Health Program. Approximately 404 lady health workers (LHWs) are also working in the district. LHWs are publicly funded; however almost half of the district does not have any LHWs appointed. After about 18-months training, a LHW is assigned to provide health education, basic birth spacing interventions and counselling, and nutrition education to about 100 houses in her catchment area near her home. LHWs earn about USD 300 a month and report to the nearby primary healthcare facility and the provincial LHW program. The LHW program has continued to expand since 1994, and currently about 125,000 LHWs cover about 60% of the country [12].

## Participant recruitment

Participants of the study were selected from 19 villages in 5 union councils of the district. These areas represent mostly Sindhi ethnic communities and are within ~10 to 20 kilometres from the main towns. We included four diverse groups of participants: 1) women who had a perinatal death in the previous 12 months, 2), family members of these women, 3) female medical officers and district health management officials, and 4) LHWs. Women who had had a fetal death (after 28 weeks of pregnancy) or an early newborn death (in the first seven days following birth), in the year preceding the interviews were considered eligible to be included in the study. Women were identified by the local LHWs who record all perinatal deaths in their catchment areas, regardless of the place of death; homes, private maternity centres, public health facilities, and these are then reported to the district health office and the LHW program. The LHWs contacted the women on the phone and helped fix the date and time for the interview. Further recruitment of women in the study was stopped when we reached data saturation and a reasonable representation of women from most prominent areas of the district was achieved.

## Data collection

Separate semi-structured IDI guides were used to interview participants for stillbirths and early newborn deaths. A separate guide was used for the FGDs with the LHWs. The guides were developed after literature review, and previous similar research experience [13]. The final versions of the IDI and FGD guides were approved after revision of several draft versions of the guides, and input by co-authors. IDI guides for the mothers and their relatives and community members contained items inquiring about their household and family situation, antenatal care (ANC) experience, their experience of stillbirth or early newborn death, and their opinion about the health services and support received during pregnancy and birth. The interview guides for the IDIs with health officials and the FGDs with LHWs included items on their perceptions about the burden of perinatal deaths in the district, and existence and use of relevant policies and actions by responsible healthcare stakeholders to prevent perinatal mortality in the district. The IDI guides were then translated into Sindhi by a professional translator and back translated by JA, who is a native Sindhi speaker.

Women and their family members were interviewed in their homes, health officials in their office hospitals, and FGDs were conducted in the basic health units. The interviews were conducted by a local Sindhi speaking female interviewer in places where participants felt comfortable such as homes. The interviewer was a local primary school teacher with previous experience of conducting surveys with international aid agencies in the province. The interviewer received intensive training and interviewing skills by JA. She was also coached on an ongoing basis using the completed interviews to improve the quality of subsequent interviews.

The interviews with health officials and FGDs were conducted by JA. Interviews were recorded and lasted between 30 minutes to an hour, and FGDs lasted approximately 45 minutes.

## Data analysis

Data were analysed using thematic analysis with both deductive and inductive coding. The transcripts were initially read for codes based on the field guide questions, and in the next step, the data were categorised into emerging themes. The IDIs and FGD data were analysed together. The interviews were transcribed verbatim in Sindhi and the translated into English by JA). The translated interview text was then read again for any errors, and initial coding was done. The coded text identified during the first reading was coded as nodes and sub-nodes in NVIVO version 11 (QSR international). Next, interview text was read and re-read and it was labelled under appropriate nodes. Finally, the node names were edited, and their hierarchy was changed by merging, and cutting and pasting them within the most appropriate themes. A framework was developed by identifying themes as they were explained by the codes. The women's discussion, of their practices during pregnancy and birth, and encounters with healthcare providers during pregnancy and birth, was used to understand their experiences about care, their perinatal losses, and the circumstances under which these losses occurred. Data from the interviews with the women were triangulated with those from the health officials and community members. We translated common terms used by the participants for their symptoms, complications, or a treatment into the most suitable medical terminology. For example, we interpreted "bleeding during pregnancy" as antepartum haemorrhage, "low blood" as anaemia, "blood injection" as injectable iron, "baby drowned after birth" as hypoxia or birth asphyxia. Extreme caution was observed in interpretation of the data by agreeing on final coding, evaluation of the analysis, and by providing feedback on several drafts of the paper by the two senior researchers who are experienced in qualitative research. We had author discussions to practice reflexivity as we considered our own personal experiences, knowledge, and beliefs. Thorough reflexivity was practiced in the analysis and interpretation phase of the study, and we challenged the first author (who belonged to the same province, spoke Sindhi and had knowledge of local healthcare system) to avoid subjectivity which may have influenced the analysis and interpretation of the data. The findings are reported based on the Consolidated Criteria for Reporting Qualitative Studies [14].

## Ethical consideration

Ethical review for the study was obtained from the ethical review committee of Health Services Academy Islamabad (F. No. 7-82/2017-IERB). Verbal consent was given by most and written by some participants after the interviewer read them the research details.

## Results

Participants included 25 women; of whom 11 had a stillbirth and 14 an early newborn death (Table 1). We also approached the family members who were available at the time of interviews and four agreed to be interviewed. The youngest mother in our study was 17 years and the oldest 35 years and their mean age was 26.5 ± 5.4 years. Most mothers did not have any formal education and were not employed, although some women worked in agriculture. Most women were multiparous (or grand multiparous). Their husbands worked either in the fields or were casual labourers. Most families had limited financial resources, including insufficient funds for out-of-pocket expenses related to their pregnancy and birth care. Six health officials were interviewed in the main town of the district; they included two medical officers, a district health officer, a medical superintendent of the district hospital, and two coordinators of

**Table 1. Description of the study participants.**

| Type of participants | Number |
|---|---|
| Mothers with stillbirth | 11 |
| Mothers with early newborn deaths | 14 |
| Father of a stillborn | 1 |
| Mother-in-law of early newborn death | 1 |
| Father of early newborn death | 1 |
| Health managers and workers | 8 |
| *Focus group discussions of Lady Health workers* | *17 people, 2 groups* |

relevant services. Two FGDs were conducted with 17 LHWs from communities surrounding two basic health units. The LHWs participating in the FGDs were from the same communities of the mothers.

We identified three interrelated but distinct themes to describe the women and their families' experience and perceptions, and health official's perspectives of factors linked with perinatal mortality: 1) poor access to care during pregnancy and birth, 2) unavailability of appropriate healthcare services, and 3) poor quality of care during pregnancy and birth.

## Poor access to care during pregnancy and birth

The most significant barriers to accessing care during pregnancy and birth for the women in our study was a lack of prompt decisions to seek skilled care around birth, delays by maternity care providers and cost of birth care services. A lack of prompt decision making about the place of birth hindered women's access to care at birth, especially since because these decisions were frequently made when the women were in labour. Many women attempted births at home with traditional birth attendants (TBAs); and when the labour did not progress, only then did they travel to a skilled birth attendant. Many of the women who birthed at home regretted this decision, because they would have preferred to birth with a skilled birth attendant.

> *"Dai (TBA) said that the baby would be born on the way to hospital, so it was better to deliver at home. She delivered in 15 minutes, but the baby got stuck halfway and died. We had never thought that we would give birth at home, we had planned to go to N (Lady Health Visitor, LHV's clinic)."*

> *(Woman with stillbirth-IDI16)*

There were some instances of home births due to a delay in access to skilled care during birth, mostly following advice from the TBAs. In retrospect, women often blamed TBAs for misleading them to believe that their babies would be born at home, and according to the women, the TBAs mishandled their labour and advised against seeking a skilled birth attendant.

> *"My (labour) pain started at three in the morning and it took until eight in the night. Baby was stuck inside (when TBA, attended the birth at home). I fought with Dai, I told her that whatever had happened, it was all because of her. She did that to me."*

> *(Woman with early newborn death-IDI23)*

Some women blamed private birth providers for causing delays in referring women to access appropriate care during labour. Such delays by the private clinics were often associated with prolonged labours and perinatal losses.

*"She (Private Dr S.) said that I would deliver in the morning and she left me there. The baby died in those pains. I had spent all night in her clinic. She did not check me. She went to sleep after saying that my baby would be born in the morning. On the next day also, she said that the baby will be born soon but it did not. After finishing (killing) the baby there, she gave me a letter for the civil hospital in Hyderabad, where they said that the baby was not moving."*

*(Woman with stillbirth-IDI20)*

A preference for private care was another reason that limited women's access to normal birth with a skilled birth attendant. Some women who had birthed at a basic health unit, reasoned that they selected a public health facility that time because they could not afford to pay for private birth care. However, their discussions implied that their previous births were at private centres because these centres were highly valued by women and their families.

*"We were poor at that time (during birth), otherwise I had been delivering with S. (Private LHV) before."*

*(Woman with stillbirth-IDI18)*

The cost was also a strong barrier for most women who were advised that they needed an elective Caesarean Section (CS). Women could not afford the CS costs in the local private hospitals which ranged between PKR20,000 (143 USD) to PKR 35,000 (250 USD). It was also common that families would borrow or sell assets to manage these costs. When referred to the civil hospital in Hyderabad to avail a free CS, the women could not even afford the cost of travel to reach the hospital, because a cost-free ambulance transfer was not available, leading these women to return to their homes to birth with TBAs.

*"We did not have money to go to (civil hospital in) Hyderabad, so we came back home. We thought that we would go in the morning, but then the (dead) baby was delivered in the midnight by Dai."*

*(Woman with stillbirth-IDI24)*

## Unavailability of appropriate healthcare services

A lack of availability of essential services such as blood transfusion, CS and newborn care was linked with multiple incidents of perinatal losses in our study. Many women were advised by their healthcare providers to receive blood transfusions due to anaemia which the women referred to as "low blood". These health providers were both private and public and various cadres of skilled birth attendants e.g midwives, nurses, and medical doctors. Since a free blood transfusion facility was not available in the public health facilities, most women could not receive a blood transfusion, as the private services was costly, and also required families to find a matching blood donor which they were unable to do.

*"We could not find it (the donor for blood group A+ve). We were poor. We did not have money"*

*(Woman with stillbirth-IDI9)*

Many women reported receiving drips and injections in place of blood transfusions mostly from private providers in the nearest towns.

*"He (Dr) said that my blood was low, and I needed blood transfusion, but we did not get it done. But we received a drip that is given in place of blood and costs 800 Rupees (5 USD)"*

*(Woman with early newborn death-IDI14)*

A lack of availability of emergency obstetric care services was a common reason for the staff in the district hospital to avoid admitting women who commonly needed a CS. These women were turned away or referred to the civil hospital in Hyderabad, or a local private hospital.

*"The district hospital doctor said that my condition was too complicated, and she could not treat me. She told us we could go to the private clinic where we usually went."*

*(Woman with stillbirth-IDI24)*

Health providers agreed that the availability of a quality CS facility in the district was important. They described that the district hospital faced many challenges related to financing the services, human resources, and equipment; thus, although the hospital had an obstetrician and an anaesthesiologist, a CS facility was not available.

*"The caesarean cases need to be done here in the district. The project people say that the hospital is complete, but how can it be complete without necessary equipment and staffing?"*

*(Health official-IDI35)*

Some mothers who underwent emergency CS at private health centres, were already in an advanced stage of labour. They described symptoms in their babies that suggested aspiration and intrapartum hypoxia. These babies often required specialised neonatal care; however, because of the unavailability of such care in the district, many were transferred by their families to the civil hospital in Hyderabad.

*"But the doctor refused me at last-minute and told that I would have operation (CS) because the baby had slow heart rate. Then I had operation however, she said that water had entered the baby and it would need two days in specialised care. She refused us treatment here (as it was not available) and we took the baby to Hyderabad where he was alive for two days."*

*(Woman with early newborn death-IDI26)*

These gaps in the availability of essential newborn care services were also recognised by senior district health officials. They highlighted that although the district hospital had an inpatient area for admitting sick children, a reasonably equipped newborn care facility was absent in the hospital. Therefore, sick newborns were referred to the civil hospital in Hyderabad, which also lacked adequate facilities. A senior health administrator described how the civil hospital was under pressure to deal with sick newborns, and this was the reason that newborn mortality was extremely high in the district.

*"The paediatrics wards in civil hospital in Hyderabad have patients (referred) from other parts of the province. If they see improvement in sick babies, they discharge them early,*

*because they are short of beds. Therefore, when patients are referred from here (district), they end up in a lot of trouble. Some babies find beds while others do not."*

*(Health official-IDI34)*

## Poor quality of care during pregnancy and birth

Women in our study described incidents which suggest that the quality of care that they received in both public and private health centres was poor. Women reported receiving insufficient information from healthcare providers about birth complications they later experienced. Some women who had complicated pregnancies including twin pregnancies, placenta praevia or abnormal presentation, blamed ANC care providers of not informing them in time that they may have a complicated birth. Women believed that if they had enough information earlier in their pregnancies, they might not have had the perinatal loss.

*" If she had told me about the problem (low fetal heart rate) and that I would not have normal delivery; we would have immediately gone for the operation (CS)."*

*(Woman with early newborn death-IDI26)*

Another aspect of lack of quality of care that the women in our study identified was the health providers' negative attitudes during interactions with the women. Some women reported incidents when they visited public health facilities for an ANC check-up that the staff were reluctant to attend them and told them that they were too busy on that day. The women perceived that they were disregarded and ignored by the staff.

*"When I went there (Basic Health Unit), they said that they were too busy on that day, they did not check it (blood test)."*

*(Woman with early newborn death-IDI12)*

LHWs also shared incidents where they accompanied pregnant women, often at their own expense to the basic health units, but the women were not attended appropriately by the public health facility staff member.

*"I brought a woman with high blood pressure to the basic health unit. The lady doctor asked me to take her to the male doctor. He said that he was not available (at that time). That is why people do not want to come here. They say that doctors tell them that they are fed up by seeing them so often."*

*(FGD2-LHWs)*

There were also many stories that suggest over-servicing through the provision of unnecessary injections, drips, and other medications predominately from private providers. Although these were inappropriate, ANC care that provided injections and drips was highly valued by the women and their families. If they did not receive these injections and drips during their pregnancy, they attributed not receiving them to be the cause of the perinatal death.

*"I think if I had received injections, drips and treatment (during ANC) then the baby could have survived."*

*(Woman with early newborn death-IDI1)*

When discussing the events surrounding their births and immediate postpartum period, some women who had birthed at basic health units and had a perinatal loss, felt that the staff did not take steps to identify the high risks to the fetus and newborns, and the immediate newborn care was poorly provided.

*"We expected that the basic health unit would do the delivery. They would have gas ($O_2$). They did not keep the boy in the gas and as soon as they removed him, the doctor went for taking shower. When my sister-in-law shouted and then they tried something to pump air in him. She (the doctor) left him like that and went away."*

*(Woman with stillbirth-IDI18)*

Quality of care issues were also indicated by incidents of perinatal loss in women who had attended private clinics. The accounts of their labour suggested substandard birth care because they were frequently treated by the unskilled assistants in the absence of the skilled birth attendants who were at the same time owners of these private clinics.

*"The assistants called the nurse and told her on phone that I was not delivering the baby. She told them to give me an injection and if I do not deliver even after that, then let me go and get the operation done. Then they gave me injection. I had a lot of pain and baby was born in five minutes, but she was dead."*

*(Woman with stillbirth-IDI3)*

Regulating private providers to improve their service quality was described by the district health authorities as challenging. Multiple public health officials expressed opinions indicating that private providers often use their influence on the district administration to avoid any accountability whenever their practice was questioned by the district health authorities.

*". . .they (private providers) know about the consequences of their malpractice so they protect themselves in that way. There have been cases when we closed a clinic and then allowed them to run the clinic after we came under pressures."*

*(Health official-IDI35)*

## Discussion

Although our participants reported a high use of ANC services, their access to a high-quality ANC and birth care was limited. Delays by families as well as health providers affected access to skilled care around birth. Financial constraints contributed to reduced access to care, especially emergency obstetric and newborn care. A lack of availability of sick newborn care, blood transfusion and CS services in the public health facilities of the district was experienced by women who had a perinatal death. Negative staff attitudes, provision of insufficient information including potential high risks during pregnancy and birth, and ignoring possible risks of perinatal morality, like high blood pressure, were the main quality of care issues experienced by the women attending public health facilities in our study. Additionally, despite a preference for private providers, the care described by the women seemed mostly substandard and negligent. Instead of receiving evidence-based care, women received unnecessary and costly injections and drips from private providers during pregnancy.

Delays in accessing skilled care at birth and subsequently birthing at home, delays in decision making for referral to appropriate care by the private health providers could have impacted perinatal losses among women in our study. Delays in seeking and receiving care are well-established risk factors of perinatal and maternal mortality [15]. Delays in taking decisions to seek care and a delay in reaching a healthcare facility contribute to newborn mortality resulting mainly from pre-term birth, birth asphyxia, and sepsis [16]. Although some women delayed their decision to attend a skilled provider and delivered with a TBAs, most delays were by private providers who failed to refer women to more appropriate care. Such delays during referrals between health facilities are linked with higher perinatal and child mortality. Our findings on delays are consistent with a study from Sierra Leone [17], which showed that such delays result from a culture of seeking guidance from TBAs, a lack of access to care at birth, and financial constraints. Another reason for delay in seeking appropriate care at birth was that women were not aware of the duration of their gestation and therefore of the due date, and therefore were not offered induction of labour at 39 weeks [18]. Appropriate and timely management of women with high-risk pregnancies and at risk newborns can reduce the risks associated with transfers between institutions [19].

Our analysis demonstrates that a lack of publicly available emergency obstetric care services led to avoidable perinatal deaths. Receiving basic and comprehensive emergency obstetric care can prevent stillbirths by 45% and 75%, respectively [20]. Timely identification of women requiring CS is linked with a reduction in perinatal mortality burden [21]. Improving CS rates in less developed countries from 1 to 10% reduces newborn mortality from 23.3 to 20.3 per 1000 live births, and maternal mortality rates from 183.1 to 118.6 per 1000 live births [22]. With adequate human resource allocation and provision of necessary equipment, surgical theatres may be enabled to offer an affordable CS service in public health facilities of most low and middle-income countries [23].

Poor availability of emergency newborn care in the public healthcare system of the district was recognised by all participants. Referral of sick newborns to specialised care needs to be < 3hours, as longer delays are associated with a doubling of the risk of death [24]. A lack of availability of an affordable ambulance transfer by the local healthcare management compounded the effects of poor service availability. Approximately a quarter of Pakistani women still prefer to birth at home [25]; therefore, identifying them and their sick newborns through local health workers and facilitating them to reach skilled emergency care would be essential to any perinatal morality prevention effort.

Cost of care impaired women's access to emergency obstetric care and newborn care. Many parents could not afford the inpatient care for their sick newborns and insisted upon ambulatory care and women who needed a CS, opted for a normal delivery because they could not afford the CS cost. Women needing a CS in our study were reluctant to travel to the civil hospital in Hyderabad when they required a CS. Travel costs and other expenses of staying in a public hospital were also not affordable, approximately 82% of women using public care and 96% using private care pay out of pocket for comprehensive emergency obstetric care services in Pakistan [26]. Access to high-quality intrapartum care has been improved by financial protection strategies resulting in significant reductions in perinatal mortality [3, 27].

Our finding that some women in our study were ignored by health facility staff is consistent with another study from Pakistan which found that 97% women experience at least one type of disrespectful or abusive behaviour by their birth attendants [28]. Negative staff attitudes and behaviour is not only linked with poor perinatal outcomes [29], but also diminishes the positive experience during pregnancy and birth, and lowers their perception of quality of care [30]. A meta-synthesis of qualitative studies identified poor staff attitudes as a reason for women's lack of service use especially of ANC [31]. In our study, women reported that ANC clinic staff

were rude and too 'busy' to see them. A shift in attitude would be required for public sector staff to provide woman centred care. In health facilities with overworked providers, a realistic approach to human resource management is required considering the harmful effects of staff burnout on patients [32]. These negative experiences in the public health facilities and perceptions of poor staff attitude towards women may also have informed women's choice to seek care from private providers or give birth at homes leading to higher risk of perinatal mortality. Apart from staff behaviour related issues, women, especially those with known conditions during pregnancy, faced other ANC quality issues such as receiving inadequate information about identified risks. ANC quality is associated with higher attendance at skilled care services around birth leading to positive pregnancy outcomes [33].

Women in our study experienced negligence during births by private providers who they blamed for not taking the necessary steps to prevent their pregnancy losses. Most of the skilled providers in our study were private lady health visitors, nurses and doctors, who offer first-line care to rural populations; yet little was known about the quality of the care they provide [34]. The incidents such as allowing unskilled assistants to attend births in their absence suggests that private providers may not be following standard procedures. Although 86% women attend some ANC and 66% give birth with skilled birth attendants, the proportion of births attended by private providers is not clear in Pakistan [35]. Research from the communities, like our study area, show that 70% of women prefer obstetric care from private providers [26]. Poor quality of private care in Pakistan has been previously described in a study showing high newborn mortality in an urban community using mostly birth care from private providers [36].

The practice of receiving various therapeutic injections and drips, from private providers was common in our study. Based on our (JA) knowledge of practices in the area, we assume that some of these injections were most likely injectable iron; however, they may have been merely vitamin-based preparations. A high use of injections and drips in pregnancy reflects practices in the general population. Health providers give approximately 731million injections (about 5.1 per person) annually in Pakistan [37]. Injections and drips are money earners for providers who charge more than double the cost. These therapeutic injections are overused, and almost half of the providers reuse syringes in Pakistan [38].

## Strengths and limitations

This is one of the few qualitative studies exploring rural women's perinatal loss within the cultural and health context in a high perinatal mortality burden country. Participation in the study was encouraging as all women we invited to participate agreed. Most of the women had a perinatal death from between one to six months before the interviews, ensuring that the data we collected were not subject to recall bias. Another strength of the study was that JA was from the study area and hence spoke the local language which gave him an advantage in accessing the participants, gaining their trust, visualising their care-seeking pathways, and interpreting their language expressions. However, most women were generally shy and cautious during the interviews, probably because they were not used to being asked their opinions about their birth related matters, and/or the topic was difficult to talk about as many women expressed grief while recalling their losses during the interviews. We also acknowledge that although the interviewer was a local woman and well oriented with the local culture and language of the area, she was not very experienced in conducting qualitative interviews. We did address this with continuous training and feedback throughout the interviews and her skills improved throughout the data collection period. We were unable to conduct a detailed review of the availability of maternal and child health services at the health facilities and could not directly

assess the quality of care provided by any type of provider which would have further validated findings of this study.

## Conclusion

This study brings to the fore the need for a context-specific comprehensive strategy to prevent perinatal mortality in Pakistan. We recommend a comprehensive strategy to ensure rural women's access to high quality ANC and birth care services. Access to an integrated referral network, linking communities, local health facilities and comprehensive obstetric and newborn care facilities, is needed for effective perinatal mortality prevention. Publicly available and free CS and newborn care services within each district would make a major impact on the perinatal mortality burden in Pakistan. Health providers' accountability to ensure their responsiveness to pregnant women during their ANC encounters and at the time of birth will ensure women and their families' trust in the services. Private maternal and child healthcare providers need to be involved and regulated to ensure quality of care around birth. Any future research exploring the role of expanding private sector care around birth will be helpful to further understand and prevent perinatal mortality in a similar context. Further research is also recommended to assess challenges to women who live in areas where they have limited support from local health workers and are at higher risk of perinatal mortality.

## Supporting information

**S1 File.**
(DOCX)

## Author Contributions

**Conceptualization:** Jamil Ahmed, Ashraful Alam, Camille Raynes Greenow.

**Data curation:** Jamil Ahmed, Saadat Khokhar, Ramesh Kumar.

**Formal analysis:** Jamil Ahmed, Ashraful Alam, Ramesh Kumar, Camille Raynes Greenow.

**Funding acquisition:** Jamil Ahmed.

**Investigation:** Jamil Ahmed, Saadat Khokhar, Sadiq Khowaja, Ramesh Kumar.

**Methodology:** Jamil Ahmed, Ashraful Alam, Saadat Khokhar, Sadiq Khowaja, Camille Raynes Greenow.

**Project administration:** Jamil Ahmed.

**Resources:** Jamil Ahmed, Camille Raynes Greenow.

**Software:** Jamil Ahmed, Camille Raynes Greenow.

**Supervision:** Ashraful Alam, Saadat Khokhar, Sadiq Khowaja, Ramesh Kumar.

**Validation:** Jamil Ahmed, Saadat Khokhar, Sadiq Khowaja.

**Visualization:** Jamil Ahmed.

**Writing – original draft:** Jamil Ahmed, Camille Raynes Greenow.

**Writing – review & editing:** Jamil Ahmed, Ashraful Alam, Saadat Khokhar, Sadiq Khowaja, Ramesh Kumar, Camille Raynes Greenow.

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
