## [Decision Letter · Decision Letter 0]

10 Dec 2019

PONE-D-19-26928

Qualitative assessment of the healthcare use experience of women with a recent perinatal mortality in Sindh, Pakistan

PLOS ONE

Dear Jamil Ahmed

Thank you for submitting your manuscript to PLOS ONE. After careful consideration, we feel that it has merit but does not fully meet PLOS ONE’s publication criteria as it currently stands. Therefore, we invite you to submit a revised version of the manuscript that addresses the points raised during the review process.

While the manuscript is presented to a high standard, the reviewers raise a number of minor points that, collectively, would improve reader understanding of methods and findings. Please also consider the suggested change in the title.  

We would appreciate receiving your revised manuscript by 8 January 2020. To enhance the reproducibility of your results, we recommend that if applicable you deposit your laboratory protocols in protocols.io, where a protocol can be assigned its own identifier (DOI) such that it can be cited independently in the future. For instructions see: http://journals.plos.org/plosone/s/submission-guidelines#loc-laboratory-protocols

We look forward to receiving your revised manuscript.

Kind regards,

Vicki Flenady

Academic Editor

PLOS ONE

2. Please include additional information regarding the interview guide used in the study and ensure that you have provided sufficient details that others could replicate the analyses. If you developed a questionnaire as part of this study and it is not under a copyright more restrictive than CC-BY, please include a copy, in both the original language and English, as Supporting Information

4. Your ethics statement must appear in the Methods section of your manuscript. If your ethics statement is written in any section besides the Methods, please move it to the Methods section and delete it from any other section. Please also ensure that your ethics statement is included in your manuscript, as the ethics section of your online submission will not be published alongside your manuscript.

5. Please include a copy of Table 2 which you refer to in your text on page 8.

6. Please upload a copy of Figure 1, to which you refer in your text on page 7. If the figure is no longer to be included as part of the submission please remove all reference to it within the text.

Reviewers' comments:

Reviewer's Responses to Questions

**Comments to the Author**

1. Is the manuscript technically sound, and do the data support the conclusions?

Reviewer #1: Yes

Reviewer #2: Yes

2. Has the statistical analysis been performed appropriately and rigorously? 

Reviewer #1: N/A

Reviewer #2: Yes

3. Have the authors made all data underlying the findings in their manuscript fully available?

Reviewer #1: Yes

Reviewer #2: Yes

4. Is the manuscript presented in an intelligible fashion and written in standard English?

Reviewer #1: Yes

Reviewer #2: Yes

5. Review Comments to the Author

Reviewer #1: 1. I think there is a need to change the title of the study. Currently, it is little confusing.

Topic could be: Exploring women’s experience of healthcare use during pregnancy and childbirth to understand factors contributing to perinatal deaths

2. Talk a little bit more about your results in abstract section. You can reduce some text from background and methods if you like.

3.I think you can further strengthen the rationale by explicitly stating the current gaps in the community-based maternal and child care services. Line no 100

4. Specify what factors, in particular, you would like to explore through this study? sociocultural? socioeconomic? religious? individual? biological? Line no 107

5. Line no 109. Within qualitative research, what design you employed to conduct this study? exploratory? descriptive? case study? etc.

6. Under the settings heading, you have not mentioned why this Tando Muhammad Khan District has been selected?

7. Can you add the interview guides as appendix?

8. I couldn't see ethical considerations of this research in the methods section. Did you took consent with study participants?

9. Regarding analysis, the FGDs and IDIs were analysed as one data set?

10. I think it would be great if you could provide a table for study participants for FGDs and IDIs. Also, how many participants in one FGD?

11. Also, the age and other demographic characteristics of participants that you have presented in results section could also be well presented in a table.

12. I think you should rephrase your themes like unavailability of the healthcare services, poor access to care and poor quality of care. The themes should give a quick understanding about the study results.

13. I think you can also make sub-themes within themes like high cost of care within poor access to care theme and so on.

14. Discussion section can be further strengthened by some good policy recommendation as in background section you stated that .. the purpose is to reorient policies and services

Reviewer #2: Reviewer’s Comments

Qualitative assessment of the healthcare use experience of women with a recent perinatal mortality in Sindh, Pakistan

First of all congratulations on a huge undertaking. To do this type of research in a developing country is extremely challenging. Obtaining Ethics committee approval, involving translation, and accessing 25 mothers in a rural community are wonderful accomplishments.

The description of your research process was clear and accurate. The participant’s comments were powerful and supported your conclusions and recommendations. The inclusion of health care providers was also important for validating what the mothers were saying. Excellent references!

A few areas might be better worded or clarified.

I am unclear as to who lady health workers are. I googled it and found out, but I wonder if a brief description of these women, with education and training details might be good.

Line 104 is unclear to me. “we found that women with a high perinatal loss in Pakistan also have low empowerment” What is women with a high perinatal loss? Do you mean women who experience perinatal loss also experience lack of power? Maybe clarify.

Line 235 Should the last word be “highly” rather than high?

Line 401 is unclear to me. “Most women in Pakistan (82% using public care and 96% using private care) pay out of pocket for comprehensive emergency obstetric care service costs (16).” Might be clearer to say 82% of those using public care…

Line 412 I am unsure of what you are saying here. “This study also showed that the effects on neonatal (and maternal) survival were much better in least developed countries” Could you clarify or explain further?

Thank you for this work. It is applicable to my work in a different developing country and I understand the great need. Your manuscript will help us continue to take steps that will impact the stillbirth rate worldwide!

6. PLOS authors have the option to publish the peer review history of their article (what does this mean?). If published, this will include your full peer review and any attached files.

Reviewer #1: Yes: Anam Feroz

Reviewer #2: Yes: Sue Steen

---

## [Author Response · Author response to Decision Letter 0]

19 Dec 2019

Reviewer #1:

 1. I think there is a need to change the title of the study. Currently, it is little confusing.

Topic could be: Exploring women’s experience of healthcare use during pregnancy and childbirth to understand factors contributing to perinatal deaths

Response: we have changed the title of the study as suggested.

2. Talk a little bit more about your results in abstract section. You can reduce some text from background and methods if you like.

Response: we have elaborated our findings in the abstract.

3.I think you can further strengthen the rationale by explicitly stating the current gaps in the community-based maternal and child care services. Line no 100

Response: we have added more information about current challenges to the maternal and child healthcare services (Introduction, Page 5, lines 120-127).

4. Specify what factors, in particular, you would like to explore through this study? sociocultural? socioeconomic? religious? individual? biological? Line no 107

Response: we have explained and elaborated the point about the objectives of this paper (Introduction, Page 6, lines 141-143).

5. Line no 109. Within qualitative research, what design you employed to conduct this study? exploratory? descriptive? case study? etc.

Response: the key word: “exploratory” is now included in the methodology section (Methods, Page 6, line 145).

6. Under the settings heading, you have not mentioned why this Tando Muhammad Khan District has been selected?

Response: we have added a line to explain the reason for selecting the district (Methods, Page 6, lines 150-151).

7. Can you add the interview guides as appendix?

We will include the interview guides, if the journal editors would like them, and that fits with in the journal guidelines.

8. I couldn't see ethical considerations of this research in the methods section. Did you took consent with study participants?

Response: ethical consideration was on title page, we have moved it to methods section in the updated version of the manuscript. Informed and written consent was taken and a statement is now included (Method, Page 10, lines 230-243).

9. Regarding analysis, the FGDs and IDIs were analysed as one data set?

Response: yes, there is this statement in the methods: “The IDIs and FGDs were analysed together”, (Methods: data analysis, page 9, line 205)

10. I think it would be great if you could provide a table for study participants for FGDs and IDIs. Also, how many participants in one FGD?

Response: We appreciate your suggestion for the table but a traditional tabular presentation of study participants in qualitative research may not be justified, as the numbers are small, and it doesn’t add anything to the manuscript as this information is clearly provided in the text. 

11. Also, the age and other demographic characteristics of participants that you have presented in results section could also be well presented in a table.

Response: thank you for your suggestion. As explained in the response to the comment number 10, we prefer to keep demographic characteristics of participants as text. We initially presented these details as a table, but after consultation between senior researchers, the table was removed.

12. I think you should rephrase your themes like unavailability of the healthcare services, poor access to care and poor quality of care. The themes should give a quick understanding about the study results.

Response: we have changed the theme titles as advised. 

13. I think you can also make sub-themes within themes like high cost of care within poor access to care theme and so on.

Response: we had sub-themes in the initial draft, however we merged them to give a bigger picture of the story that our research tells in response to the research questions. Subthemes did not add to the data analysis nor did they assist with interpretation and there was not enough data to make them substantive, therefore we would prefer to go only with main themes.

14. Discussion section can be further strengthened by some good policy recommendation as in background section you stated that .. the purpose is to reorient policies and services

Response: we have included a policy recommendation in the conclusion section (Discussion, Page 26, line 571-575).

Reviewer #2: 

Qualitative assessment of the healthcare use experience of women with a recent perinatal mortality in Sindh, Pakistan

1.First of all congratulations on a huge undertaking. To do this type of research in a developing country is extremely challenging. Obtaining Ethics committee approval, involving translation, and accessing 25 mothers in a rural community are wonderful accomplishments.

The description of your research process was clear and accurate. The participant’s comments were powerful and supported your conclusions and recommendations. The inclusion of health care providers was also important for validating what the mothers were saying. Excellent references!

Response: the comments are extremely encouraging. Thank you.

2.A few areas might be better worded or clarified.

I am unclear as to who lady health workers are. I googled it and found out, but I wonder if a brief description of these women, with education and training details might be good.

Response: we have added details about LHWs in the methods (Methods, Page 7, lines 159-165).

3.Line 104 is unclear to me. “we found that women with a high perinatal loss in Pakistan also have low empowerment” What is women with a high perinatal loss? Do you mean women who experience perinatal loss also experience lack of power? Maybe clarify.

Response: we have clarified the sentence and elaborated it further. 

4.Line 235 Should the last word be “highly” rather than high?

Response: we have edited the word as suggested (Results, Page 13, line 288).

5.Line 401 is unclear to me. “Most women in Pakistan (82% using public care and 96% using private care) pay out of pocket for comprehensive emergency obstetric care service costs (16).” Might be clearer to say 82% of those using public care…

Response: we have edited the sentence as: “About 82% women using public care and 96% using private care pay out of pocket for comprehensive emergency obstetric care service costs in Pakistan.” We hope that it is clearer now (Discussion, Page 21, line 454-456).

6.Line 412 I am unsure of what you are saying here. “This study also showed that the effects on neonatal (and maternal) survival were much better in least developed countries” Could you clarify or explain further?

Response: we have elaborated the text with more explanation (Discussion, page 21 and 22, lines 466-468).

7.Thank you for this work. It is applicable to my work in a different developing country and I understand the great need. Your manuscript will help us continue to take steps that will impact the stillbirth rate worldwide!

Response: thanks for the kind words, this is immensely motivating, and your words further strengthen my passion for working towards saving babies’ lives.

---

## [Decision Letter · Decision Letter 1]

9 Apr 2020

PONE-D-19-26928R1

Exploring women’s experience of healthcare use during pregnancy and childbirth to understand factors contributing to perinatal deaths in Pakistan: a qualitative study

PLOS ONE

Dear Dr. Ahmed,

Thank you for submitting the revised version of your manuscript to PLOS ONE. After a second round of review, we feel that it does not fully meet PLOS ONE’s publication criteria yet. Therefore, we invite you to submit a revised version of the manuscript that addresses the points raised during the second review process.

We would appreciate receiving your revised manuscript by April 20th 2020 11:59PM. To enhance the reproducibility of your results, we recommend that if applicable you deposit your laboratory protocols in protocols.io, where a protocol can be assigned its own identifier (DOI) such that it can be cited independently in the future. For instructions see: http://journals.plos.org/plosone/s/submission-guidelines#loc-laboratory-protocols

We look forward to receiving your revised manuscript.

Kind regards,

Sara Ornaghi, M.D., Ph.D.

Academic Editor

PLOS ONE

Additional Editor Comments (if provided):

The topic of this manuscript is of utmost importance. However, manuscript's intelligibility is limited by the presence of grammar mistakes throughout the document. For instance, meaning of sentences in Line 126-128 and 132-136 is difficult to understand as they are currently written. I'd suggest to have the manuscript revised by an English-proficient individual. Additional minor comments:

- Line 114: ‘other high perinatal burden countries’. I suppose the authors mean ‘other high perinatal mortality-burden countries’ as specified in line 119. Please correct.

- Line 123: ‘..management of server infections’. I suppose the authors mean severe. Please correct.

- Line 193: please specify the occupation of the local Sindhi speaking interviewer at the time of the study and the type of training she received.

Reviewers' comments:

Reviewer's Responses to Questions

**Comments to the Author**

1. If the authors have adequately addressed your comments raised in a previous round of review and you feel that this manuscript is now acceptable for publication, you may indicate that here to bypass the “Comments to the Author” section, enter your conflict of interest statement in the “Confidential to Editor” section, and submit your "Accept" recommendation.

Reviewer #3: (No Response)

2. Is the manuscript technically sound, and do the data support the conclusions?

Reviewer #3: Partly

3. Has the statistical analysis been performed appropriately and rigorously? 

Reviewer #3: No

4. Have the authors made all data underlying the findings in their manuscript fully available?

Reviewer #3: No

5. Is the manuscript presented in an intelligible fashion and written in standard English?

Reviewer #3: Yes

6. Review Comments to the Author

Reviewer #3: The authors performed a qualitative research on the causative factors of perinatal deaths in a rural area in Pakistan, interviewing women and families involved, as well as health care officers and lady health workers.

I agree with the second reviewer on emphasizing the novelty and the effort of giving voice to women of this area, underlining social and economical problems of that society.

However, as stated by Reviewer1, the manuscript needs to be reviewed in order to sound more scientific, although maintaining its qualitative nature.

To do this, my suggestions are the followings (mainly in agreement with Reviewer 1):

• I suggest to add the interview guides as appendix: it would be useful for the scientific community in order to reproduce the same methodology for other qualitative studies on perinatal death.

• I think it would be useful to provide a table for study participants for FGDs and IDIs, although the numbers are small.

• Demographic data of participants are mandatory, presented in a table (age, BMI, parity, ...).

Other minor changes:

• Introduction line 105: I would abbreviate in "Perinatal mortality is a challenge in developing countries..."

• Introduction line 108: I would add "compared to 4 per 1000 births in high-income countries".

• Results line 256: I would use "delivered" rather than "birthed".

• Concerning grammar, pay attention to the use of articles: sometimes you can substitute "the women" with just "women". Please correct it in the manuscript.

• Try to be more concise in the Discussion.

7. PLOS authors have the option to publish the peer review history of their article (what does this mean?). If published, this will include your full peer review and any attached files.

Reviewer #3: Yes: Annalisa Inversetti

---

## [Author Response · Author response to Decision Letter 1]

20 Apr 2020

20 April 2020

Subject: Response to reviewers: PONE-D-19-26928R1

Dear Editor

We appreciate the second round of comments, which have helped improve our article. The article is now updated following the comments in the second review. In the updated version of the article, the line numbers, where reviewers had identified issues, are now changed because of the text editing. Where applicable, we have now identified sections, and page and line numbers for changes.

Additional Editor Comments:

The topic of this manuscript is of utmost importance. However, manuscript's intelligibility is limited by the presence of grammar mistakes throughout the document. For instance, meaning of sentences in Line 126-128 and 132-136 is difficult to understand as they are currently written. I'd suggest having the manuscript revised by an English-proficient individual. Additional minor comments:

Response: the article is thoroughly revised and checked for grammar mistakes.

- Line 114: ‘other high perinatal burden countries’. I suppose the authors mean ‘other high perinatal mortality-burden countries’ as specified in line 119. Please correct.

Response: we have corrected the specified errors and checked rest of the section for any additional errors (Introduction, Page 4, lines 78).

- Line 123: ‘..management of server infections’. I suppose the authors mean severe. Please correct.

Response: we have corrected the specified error (Introduction, Page 4, lines 87).

- Line 193: please specify the occupation of the local Sindhi speaking interviewer at the time of the study and the type of training she received.

Response: we have added details about the occupation, experience of the interviwer and the type of training she received for the data collection (Methodology, Pages 7-8, lines 156-162).

Reviewer #3:

The authors performed a qualitative research on the causative factors of perinatal deaths in a rural area in Pakistan, interviewing women and families involved, as well as health care officers and lady health workers. I agree with the second reviewer on emphasizing the novelty and the effort of giving voice to women of this area, underlining social and economical problems of that society. However, as stated by Reviewer1, the manuscript needs to be reviewed in order to sound more scientific, although maintaining its qualitative nature. To do this, my suggestions are the followings (mainly in agreement with Reviewer 1):

• I suggest to add the interview guides as appendix: it would be useful for the scientific community in order to reproduce the same methodology for other qualitative studies on perinatal death.

Response: As suggested, we have included the interview guides as appendix (Apendix-1).

• I think it would be useful to provide a table for study participants for FGDs and IDIs, although the numbers are small.

Response: we have included table-1 for the study participants.

• Demographic data of participants are mandatory, presented in a table (age, BMI, parity, ...).

Response: We have presented the demographic data of the participants in the results. This includes their mean age (SD), minimum and maximum age of the participants, and commented on the parity. Body Mass Index or parity or any other quantitative variables were not measured systemically because they were not part of our aims for the present study. 

Other minor changes:

• Introduction line 105: I would abbreviate in "Perinatal mortality is a challenge in developing countries..."

Response: As a general rule, we have avoided using abbreviations since most terms are not repeated very frequently.

• Introduction line 108: I would add "compared to 4 per 1000 births in high-income countries”. 

Response: We have added data about high income countries as advised (Introduction: Page 3, Lines 71-73)

• Results line 256: I would use "delivered" rather than "birthed".

Response: We have replaced "birthed" with "delivered" at places where the word was used.

• Concerning grammar, pay attention to the use of articles: sometimes you can substitute "the women" with just "women". Please correct it in the manuscript.

Response: We have removed unnecessary articles and made other grammar corrections.

• Try to be more concise in the Discussion.

Response: We have made changes in the discussion to make the section more concise. Any detail which was deemed extra is now removed.

---

## [Decision Letter · Decision Letter 2]

23 Apr 2020

Exploring women’s experience of healthcare use during pregnancy and childbirth to understand factors contributing to perinatal deaths in Pakistan: a qualitative study

PONE-D-19-26928R2

Dear Dr. Ahmed,

We are pleased to inform you that your manuscript has been judged scientifically suitable for publication and will be formally accepted for publication once it complies with all outstanding technical requirements.

With kind regards,

Sara Ornaghi, M.D., Ph.D.

Academic Editor

PLOS ONE

Additional Editor Comments (optional):

Reviewers' comments:

Reviewer's Responses to Questions

**Comments to the Author**

1. If the authors have adequately addressed your comments raised in a previous round of review and you feel that this manuscript is now acceptable for publication, you may indicate that here to bypass the “Comments to the Author” section, enter your conflict of interest statement in the “Confidential to Editor” section, and submit your "Accept" recommendation.

Reviewer #3: All comments have been addressed

2. Is the manuscript technically sound, and do the data support the conclusions?

Reviewer #3: Yes

3. Has the statistical analysis been performed appropriately and rigorously? 

Reviewer #3: Yes

4. Have the authors made all data underlying the findings in their manuscript fully available?

Reviewer #3: Yes

5. Is the manuscript presented in an intelligible fashion and written in standard English?

Reviewer #3: Yes

6. Review Comments to the Author

Reviewer #3: (No Response)

7. PLOS authors have the option to publish the peer review history of their article (what does this mean?). If published, this will include your full peer review and any attached files.

Reviewer #3: Yes: Annalisa Inversetti

---

## [Editor Report · Acceptance letter]

28 Apr 2020

PONE-D-19-26928R2 

Exploring women’s experience of healthcare use during pregnancy and childbirth to understand factors contributing to perinatal deaths in Pakistan: a qualitative study 

Dear Dr. Ahmed:

I am pleased to inform you that your manuscript has been deemed suitable for publication in PLOS ONE. Congratulations! Your manuscript is now with our production department. 

With kind regards,

on behalf of

Dr. Sara Ornaghi 

Academic Editor

PLOS ONE